# Current Understanding of Molecular Phase Separation in Chromosomes

**DOI:** 10.3390/ijms221910736

**Published:** 2021-10-04

**Authors:** Je-Kyung Ryu, Da-Eun Hwang, Jeong-Mo Choi

**Affiliations:** 1Department of Biological Sciences, KAIST, Daejeon 34141, Korea; 2Department of Chemistry, Pusan National University, Busan 46241, Korea; hde302@pusan.ac.kr

**Keywords:** biomolecular phase separation, bridging-induced phase separation, intrinsically disordered proteins, multivalent DNA-binding proteins, stickers-and-spacers framework, compartments, cohesin, chromosomes

## Abstract

Biomolecular phase separation denotes the demixing of a specific set of intracellular components without membrane encapsulation. Recent studies have found that biomolecular phase separation is involved in a wide range of cellular processes. In particular, phase separation is involved in the formation and regulation of chromosome structures at various levels. Here, we review the current understanding of biomolecular phase separation related to chromosomes. First, we discuss the fundamental principles of phase separation and introduce several examples of nuclear/chromosomal biomolecular assemblies formed by phase separation. We also briefly explain the experimental and computational methods used to study phase separation in chromosomes. Finally, we discuss a recent phase separation model, termed bridging-induced phase separation (BIPS), which can explain the formation of local chromosome structures.

## 1. Introduction

The various components of cells (especially eukaryotic cells) are organized both spatially and temporally for efficient functioning; membrane-bound organelles are examples of spatiotemporal compartmentalization. However, other types of organelles exist that lack a membrane structure, known as membraneless organelles [1], and include: nucleoli for ribosomal synthesis in the nucleus [2], centrosomes for microtubule nucleation [3], Cajal bodies for the synthesis of spliceosomes [4], and stress granules for modulation of the stress response [5]. Although these organelles do not enclose their components within a membrane, they do not simply mix with their surroundings. Recent studies have found that demixing occurs spontaneously via liquid-liquid phase separation (LLPS) [6,7,8,9,10], a phenomenon known in physics and chemistry for more than a century. Demixing behavior occurs in a multi-component system when the energy gain for demixing is greater than the entropic loss for demixing. A good example is a typical water-oil system; water-oil mixing results in the formation of unfavorable water-oil molecular interactions, which exceeds the entropic penalty of demixing. Hence, such a system favors demixing under ambient conditions.

In 2009, Brangwynne and colleagues published a pioneering study in this field [11], which showed the liquid-like properties of P granules, a type of membraneless organelle in *C. elegans*. P granules exchange their components with the cytoplasm and exhibit fusion, dripping, and wetting behaviors. The authors also estimated the viscosity and surface tension of the granules. Subsequently, the material properties and biological implications of membraneless organelles have attracted significant interest [12,13]; a membraneless organelle can recruit specific molecules, whose local concentration becomes significantly higher than the cytosol concentration. As the concentration determines the reaction rate, the membraneless organelle can serve as a reaction center of the recruited molecules. In addition, because of their liquid-like nature, membraneless organelles allow the rapid arrangement of specific molecules upon perturbations such as temperature change; cells can use this mechanism to respond rapidly to an abrupt change of the environment. LLPS is involved in various biological processes, such as immune signaling [14], miRISC assembly [15], autophagy [16], nucleolus formation [17], stress granule assembly [18], transcriptional condensate assembly [19], and cohesin cluster formation [20].

It has also been suggested that phase separation drives chromosome organization and various genome-related biological functions [21,22]. DNA, which carries the genetic information of a cell, is densely packed in the nucleus. The efficient packing of DNA from a stretched, meters-long chain into a micrometer-scale structure is accomplished by chromatin, which is a molecular complex of DNA, protein, and RNA. Chromatin can be divided into two compartments, A and B, according to the gene content and location, and chromatin compartmentalization is believed to be driven by phase separation [23,24]. In addition, membraneless condensates form inside the nucleus, called *nuclear condensates* or *nuclear bodies* [25], whose formation and regulation can be explained by LLPS [22] (Figure 1). In this review, we highlight recent advances in the contemporary understanding of phase separation in the nucleus, where phase separation involves the extremely long heteropolymer, DNA, for chromosome organization, and DNA-related biological functions.

## 2. Principles of Phase Separation

### 2.1. Basic Models of Phase Separation

Consider two types of molecules, X and Y, in a test tube. If homotypic interactions (X-X and Y-Y) are more favorable than heterotypic interactions (X-Y), the system energetically prefers the two components to separate (phase separation). Meanwhile, entropy always drives the system towards mixing. Hence, there is a “tug of war” between the two driving forces, energy and entropy, and the molecular details determine whether phase separation occurs under the given experimental conditions (temperature, concentration, salt condition, etc.). A phase diagram is utilized to summarize the conditions of phase separation for the system of interest (Figure 2).

The Bragg-Williams model [26] describes the phase separation of two-component systems. For component X with a volume fraction of $\phi_{X}$ and component Y with a volume fraction $\phi_{Y}$ ($\phi_{X}+\phi_{Y}=1)$, the model predicts the molar mixing free energy of the system, $\Delta F_{\mathrm{mix}}$, as:(1)$$\frac{\Delta F_{\mathrm{mix}}}{RT}=\phi_{X}\ln\phi_{X}+\phi_{Y}\ln\phi_{Y}+\chi_{\mathrm{XY}}\phi_{X}\phi_{Y}.$$
where R is the gas constant and T is the absolute temperature. Here, $\chi_{\mathrm{XY}}$ is the exchange parameter that quantifies the average difference between the homotypic and heterotypic interactions of X and Y:(2)$$\chi_{\mathrm{XY}}=\frac{z}{RT}\left(w_{\mathrm{XY}}-\frac{w_{\mathrm{XX}}+w_{\mathrm{YY}}}{2}\right)$$
where $w_{ij}$ indicates the two-body interaction energy between molecules of types i and j, and z is the coordination number. If $\chi_{\mathrm{XY}}>2$, the strength of the homotypic interactions exceeds that of the heterotypic interactions to counterbalance the entropic effect, and phase separation occurs over a range of concentrations (Figure 2A).

To increase the tendency toward phase separation, unit molecules can be connected covalently to construct multimers (oligomers and polymers). Each multimer can simultaneously interact with multiple counterparts, which effectively reduces the entropic cost. For a multimer-solvent system, the molar mixing free energy of the Bragg-Williams model is generalized (known as the Flory-Huggins model [27,28]) as
(3)$$\frac{\Delta F_{\mathrm{mix}}}{RT}=\frac{\phi_{X}}{M}\ln\phi_{X}+\phi_{Y}\ln\phi_{Y}+\chi_{\mathrm{XY}}\phi_{X}\phi_{Y}$$
where M is the number of binding units (*valence*) in each multimer. With this modification, the phase separation territory in the phase diagram can be markedly expanded (Figure 2B). The model can be further generalized to multicomponent systems [29,30].

The phase separation of multimers is coupled to another type of transition: networking transition, also known as *percolation*. Each multimer has multiple binding units; two multimers are (at least transiently) connected by the formation of a physical bond between the binding units of each multimer. Increasing the concentration of multimers increases the fraction of connected multimers, and at a certain threshold concentration, a large network structure emerges abruptly. This transition is called percolation. The experimental conditions for networked and unnetworked systems can be depicted using a phase diagram (Figure 2C, blue dashed line). The Flory-Stockmayer model [31,32] was the first model to investigate percolation; it concluded that for a multimer with a valence of M, the probability p for each binding unit to form a bond must exceed the threshold value to generate a system-spanning network. Because p<1 for transient interactions, monomer and dimer systems (M≤2) cannot undergo percolation [33].
(4)$$p_{c}=\frac{1}{M-1}$$

At temperatures below the critical temperature $T_{c}$ (above which entropy disrupts phase separation), three different transition concentrations can be designated on the phase diagram (see Figure 2C). As the multimer concentration increases, the saturation concentration ($c_{\mathrm{sat}}$) is reached in the system, after which the two phases are separated. Subsequently, the percolation concentration ($c_{\mathrm{perc}}$) is reached, which divides unnetworked and networked systems. Because the spatial proximity of multimers is driven by bond formation, the percolation concentration is coupled to the saturation concentration [34,35]. Finally, at the droplet concentration ($c_{\mathrm{drop}}$), the system re-enters the one-phase region.

If the multimer concentration, c, is between $c_{\mathrm{sat}}$ and $c_{\mathrm{drop}}$, the solute multimers are either in the dilute (whose concentration is $c_{\mathrm{sat}}$) or in the dense phase (whose concentration is $c_{\mathrm{drop}}$). The amounts of molecules in the two phases are governed by the conservation of molecule number and volume, and are determined by the following rule (called the *lever*
*rule*):(5)$$N_{\mathrm{dilute}}:N_{\mathrm{dense}}=\left(c_{\mathrm{drop}}-c\right):\left(c-c_{\mathrm{sat}}\right)$$
where $N_{\mathrm{dilute}}$ and $N_{\mathrm{dense}}$ indicate the amounts of solute multimers in the dilute and dense phases, respectively. The lever rule states that: (1) if $c=c_{\mathrm{sat}}$, all solutes are in the dilute phase; (2) if $c=c_{\mathrm{drop}}$, all solutes are in the dense phase; and (3) if c is between $c_{\mathrm{sat}}$ and $c_{\mathrm{drop}}$, there are a finite number of solutes in each phase, and as c nears $c_{\mathrm{drop}}$, more solute molecules move from the dilute phase to the dense phase. This is reflected in the observations that after crossing $c_{\mathrm{sat}}$, the size and number of dense-phase droplets increases as the solute concentration increases.

### 2.2. Stickers-and-Spacers Framework

Proteins are the essential driver of biomolecular phase separation, and their roles and mode of action in LLPS have been extensively studied. In this section, we discuss a simple conceptual framework that can explain the phase behaviors of proteins. The framework is useful in understanding biomolecular LLPS and can be extended further to other multimer systems. Two representative types of protein are known to undergo phase separation. Multi-domain proteins possess well-defined folded domains connected by disordered linkers. Several multi-domain protein systems have been reported to exhibit phase separation behavior [36,37,38]. A more prominent group is comprised of intrinsically disordered proteins (IDPs), which lack well-defined three-dimensional structures, even under physiological conditions [39]. Many phase separation systems identified in vivo contain significant portions of intrinsically disordered regions (IDRs) [40]. DNA and RNA, an important group of biomolecules in living cells, can also participate in intracellular phase separation [41,42,43].

Multi-domain proteins and IDPs can be analyzed conceptually using the so-called *stickers-and-spacers framework* [8,44]. Inspired by theories of associative polymers [34,45], this framework partitions the target protein into two regions: molecular fragments responsible for chain-chain interactions (stickers), and the rest of the molecule, which is considered relatively inert (spacers). Spacers are assumed to modulate chain properties; however, their influence on chain-chain interactions is significantly weaker than that of stickers. In the case of multi-domain proteins, the partitioning is straightforward: interacting domains act as stickers, while disordered linkers act as spacers. IDP systems are more complicated, but experiments have shown that many systems can identify a set of amino acids that behave like stickers [44,46,47]. Notably, the dichotomy between multi-domain proteins and IDPs is fairly arbitrary, considering the spectrum of interactions involving amino acids, short linear motifs, peptides, and proteins. For example, short linear motifs on an IDP can interact with domains on a multi-domain protein to drive phase separation, in which the short linear motifs and the folded domains act as stickers, despite their different lengths. The stickers-and-spacers framework does not require the dichotomy.

An IDP is considered as a prototypical model containing multiple types of stickers. There are 20 different canonical amino acids; considering post-translational modifications, the number of different types of IDP monomers in vivo far exceeds 20. Thus, it may appear unusual that only a handful of amino acids dictate the phase behavior of the whole protein chain, as there is a jumble of multiple different interactions. This apparent paradox can be answered by a mean-field model of multi-sticker systems, where each multimer contains different types and numbers of stickers [48]. According to this model, the percolation concentration, which can be used as a proxy for the saturation concentration, is as follows:(6)$$c_{\mathrm{perc}}\approx \frac{1}{\sum_{i}v_{ii}e^{-\beta w_{ii}}s_{i}^{2}+2\sum_{i,j>i}v_{ij}e^{-\beta w_{ij}}s_{i}s_{j}}$$
where i and j are indices for sticker types, $s_{i}$ is the number of stickers of type i in each multimer, $v_{ij}$ is the bond volume, in which a pair of stickers is spatially constrained after bond formation, $\beta =1/k_{B}T$, and $k_{B}$ is the Boltzmann constant. Each term $v_{ij}e^{-\beta w_{ij}}s_{i}s_{j}$ in the denominator indicates the contribution of each sticker pair (i,j); the contributions are additive and independent. Note that the term contains both intrinsic and extrinsic properties of a sticker pair: the bond volume $v_{ij}$ and the interaction energy $w_{ij}$ are invariable for a given sticker pair, while the numbers of stickers, $s_{i}$ and $s_{j}$, can be modulated by mutagenesis. If the contribution of a certain sticker pair (p,q) dominates the denominator, the apparent percolation concentration becomes:(7)$$c_{\mathrm{perc}}\propto \frac{1}{v_{pq}e^{-\beta w_{pq}}s_{p}s_{q}}$$

In this system, only monomer types p and q play the role of stickers, and other monomer types are considered spacers. Hence, depending on the amino acid composition of an IDP, the identity of stickers can differ between systems. Typical sticker interactions involve cation-anion interactions, π–cation, and π–π interactions [8].

### 2.3. Multi-Component Systems

Different types of biomolecular condensates have distinct compositions. For example, the proteomes and interactomes of P-bodies and stress granules only marginally overlap [49]. Then, for a system containing multiple components, how many distinct condensates can we have? The generalized Flory-Huggins model was recently applied to address this question; the maximum number of distinct condensates was found to increase much faster than the number of components [30].

Biomolecular condensates consist of hundreds or thousands of different types of biomolecules. Do they all contribute to the formation of condensates, or is there a subset of essential players in condensate formation? The latter seems to be the case in most systems, and the essential drivers are termed *scaffolds*. Typically, scaffolds are defined as molecules that can form droplets when isolated in vitro (to be rigorous, the removal of scaffold molecules from in vivo condensates must be shown to interrupt phase separation). The other molecules are recruited to condensates by their interactions with the scaffolds and are termed *clients* [50]. Although clients are not necessary for the formation of condensates, they can modulate the properties of condensates [51]. Recruitment leads to the non-uniform distribution of client molecules inside the condensates, as they tend to remain around the scaffolds [52].

### 2.4. Microphase Separation

Phase separation can be used to generate and modulate the structure of a gigantic macromolecule. From the perspective of polymer physics, chromosomes can be considered large polymers, resembling beads on a string. Each “bead” is slightly different, and the interactions between beads are complex. According to polymer models, if a polymer consists of building blocks with different interactions, it can undergo *microphase separation*, the formation of distinct microdomains enriched in different types of building blocks because of intrachain phase separation [53]. Depending on the fraction of each building block and their interaction strengths, the polymer system can exhibit diverse mesoscopic morphologies [54]. In the stickers-and-spacers framework, the stickers gather to form microdomains, and the spacers provide sticker connectivity, which prevents the perfect segregation of microdomains. The microphase separation model is appealing because it explains local and global structure formation, and the regulation of chromatin, although it may not be the only mechanism for genome folding [55].

## 3. Phase Separation in a Nucleus

Phase separation seems to have diverse roles in the cell nuclei. For example, chromosome organization, transcription regulation, DNA damage repair, and RNA splicing are related to phase separation (Figure 1). An important feature of these processes is that long DNA molecules are involved in the formation of their corresponding biomolecular condensates. In this section, we discuss a few examples of these processes and their biophysical properties.

### 3.1. Chromatin Compartmentalization

Interphase chromosomes are segregated into two distinct compartments. The transcriptionally active, gene-rich form of chromatin is called *euchromatin*, and the transcriptionally inactive form is called *heterochromatin* (Figure 1, red and blue denoting euchromatin and heterochromatin, respectively) [56,57,58,59,60,61]. Compartmentalization seems to be driven by the phase separation of some proteins, such as heterochromatin protein 1 alpha (HP1α), a protein enriched in heterochromatin. Recent studies have shown that HP1α induces liquid droplet formation, and droplet formation tightly compacts DNA, supporting a role for the phase separation of HP1α in chromosome organization [23,24].

The two compartments were originally defined by Emil Heitz (1892–1965) about a century ago using a DNA-staining method [62,63]. Because of the different DNA densities of the two compartments, Heitz differentiated the densely stained, condensed form of heterochromatin from the lightly stained, decondensed form of euchromatin. It was found later that nucleosomes are sparsely distributed in euchromatin and densely distributed in heterochromatin, and that this induces higher accessibility of DNA to transcriptional factors in the former than in the latter [21]. The inaccessibility of heterochromatin might be explained by HP1α driving phase separation, as it can tightly compact DNA via transient interactions between HP1α and specific histone markers, such as H3K9me3 or H3K27me3 [64,65]. However, the detailed molecular mechanism underlying chromatin compartmentalization is not clearly understood.

Microphase separation has been proposed to explain the segregation of heterochromatin and euchromatin, as chromatin can be considered as a copolymer consisting of alternatively localized euchromatin and heterochromatin, forming distinct microdomains in two compartments [53]. Chromatin contact analysis (high-throughput chromosome conformation capture, or Hi-C, see Section 4) on interphase chromosomes was shown to present checkerboard contact patterns [56,57,58,59,60,61], indicating that the two types of chromatins are spatially segregated and that each type of chromatin prefers to interact with the same type [57,66]. Eigenvector deconvolution analysis of the experimental data revealed two principal compartments, termed A and B, corresponding to euchromatin and heterochromatin, respectively.

Epigenetic analyses, such as chromatin immunopreciptation with high-throughput sequencing (ChIP-seq) and ATAC-seq, can also be used to identify chromatin domains, since the two types of chromatins are marked with different types of epigenetic modifications. Histones of euchromatin are marked by H3K4me3, H3K27ac, H4K8ac, and H4K16ac, whereas those of heterochromatin are marked by H3K9me3 or H3K27me3 [64,65]. Epigenetic analysis revealed that euchromatin and heterochromatin regions alternatively localize along the chain of each chromosome [67,68,69,70], which also supports the microphase separation of a large polymeric chromosome.

### 3.2. Nucleolus

The nucleolus is a membraneless organelle in each nucleus, which is formed by LLPS of nucleolar proteins [71]. The nucleolus provides a site for ribonucleoprotein particle assembly, primarily for ribosome biogenesis, and it also serves other processes to maintain cell homeostasis [17]. In mammalian cells, the nucleolus comprises an interesting, layered structure with three functionally and compositionally distinct subcompartments: the fibrillar center (FC), the dense fibrillar component (DFC), and the granular component (GC). The FC, the innermost layer, initiates ribosome biogenesis, and as preribosomal and ribosomal molecular components diffuse from FC to DFC to GC, the ribosome is assembled in an orderly manner through a complex and dynamic process [72].

The nucleolus is an example of the scaffold-client model. Among hundreds of different biomolecules within a nucleolus [73], only a few proteins correspond to the formation of droplets as well as layered structures. Fibrillarin (FBL) is a protein that participates in the processing of ribosomal RNA and is enriched in DFC. Nucleophosmin (NPM1) is a protein associated with nucleolar ribonucleoprotein structures and is abundant in GC. A mixture of FBL and NPM1 was shown to reproduce phase separation in vitro and generate two-layer droplets, similar to the DFC-GC structure [74].

The molecular structures of FBL and NPM1 illustrate the stickers-and-spacers architecture. Both FBL and NPM1 contain IDRs, with FBL displaying Arg-rich domains and NPM1 displaying acidic tracts, which consequently interact via electrostatic interactions. NPM1 forms a pentamer that provides multivalency. In addition, FBL and NPM1 can bind to RNA via their RNA-binding domains, permitting additional transient interactions [74]. Indeed, RNA has been shown to promote nucleation and lower saturation concentrations [6,75]. Therefore, these molecular features of the nucleolus show that the LLPS model provides a simple and powerful explanation of the structural maintenance and function of the nucleolus [17].

### 3.3. Transcription Condensates

Recent studies have shown that transcription factors (TFs) and RNA induce the formation of transcriptional condensates via LLPS, which contain clusters of multiple enhancers (*super-enhancers*) [76,77]. This hypothesis is supported by the dynamic interaction of TF compartments with RNA polymerase II (Pol II) clusters [76,77]. To form transcriptional condensates, TFs bind to various *cis*-regulatory DNA elements (e.g., promoters and enhancers) and stimulate the transcription of active genes in proximity [78], facilitating the precise control of gene expression. Enhancers and promoters provide multiple binding sites for TFs, which are needed to concentrate TFs and form transcriptional condensates. In addition, transcriptional condensates present a liquid property; two different transcriptional condensates can be merged, and fluorescence recovery after photobleaching (FRAP, see Section 4) analysis revealed a clear exchange of TF molecules between the background and the condensates [19].

The structural features of typical TFs can explain how TFs induce LLPS. Typical TFs possess IDRs that can weakly interact with those of cofactors, and these multivalent interactions can induce dynamic assembly formation and be controlled by post-translational modification. Generally, TFs have stable structured domains for selective DNA/RNA binding, which provide additional weak interactions [79]. For example, FUS, EWSR1, and TAF15, known as the FET family, are mostly disordered and capable of binding to RNA molecules [80]. These are well-known model systems for phase separation in vitro [81,82]. The TFs interact with the intrinsically disordered C-terminal domain of Pol II, and this C-terminal domain is key to the formation of large spherical droplets, which possess a liquid property in living cells [83] even at endogenous expression levels [19,84].

### 3.4. Viral Genome Organization

Like phase separation of eukaryotic nuclear proteins and prokaryotic nucleoid proteins, phase separation of viral proteins is involved in the cellular processes of virus [85,86,87]. For example, RNA viruses, such as respiratory syncytial virus (RSV), vesicular stomatitis virus (VSV), and coronaviruses, appear to replicate themselves in *viral inclusion bodies*, membraneless condensates formed by phase separation, in host cells [85,86,88,89,90]. Moreover, several studies on coronaviruses have shown that the assembly of viral capsids and genomes occurs in dynamic cytoplasmic foci formed by phase separation [91,92], suggesting that phase separation plays a role in the replication and packaging of coronaviruses. Coronaviruses contain a relatively long 30 kbp single-stranded RNA genome and are compacted in a viral particle in a highly specific manner by excluding host RNA and many subgenomic RNAs [93]. In particular, the nucleocapsid protein (N-protein) of SARS-CoV-2 drives viral RNA genome packaging using LLPS, which is mediated by interactions between specific viral RNA sequences and multivalent RNA-binding domains and IDRs of the viral proteins [87,94,95,96,97,98,99]. Some specific RNA sequences interact with the N-proteins for LLPS, and this seems to ensure that the viral RNA is not entangled with other long cellular RNA molecules [100,101]. LLPS studies on viruses provide novel perspectives on how the composition of RNA determines its packaging into a small viral particle.

## 4. Technical Approaches to Study Phase Separation in Chromosome

Different biophysical and biochemical approaches have been employed to study intracellular phase separation [13,102,103,104]. One approach for investigating intracellular phase separation is to reconstitute biomolecular condensates in vitro, using minimal and essential components, and explore the physical properties of the condensates. In-vitro reconstitution can provide detailed information on how biomolecules interact to form a biomolecular condensate; typical chemical tools can be utilized here. Although in-vitro studies can provide detailed biophysical information on the condensates, the data should be confirmed using live-cell experiments to enable biologically relevant conclusions to be drawn. Live-cell imaging is used widely to monitor condensates and study the characteristics of condensates inside a cell [19,105]. Conversely, genomic analyses, such as sequencing techniques and Hi-C, have been used to study chromosome organization, where phase separation can play a role, as discussed earlier [106]. In addition, computer simulations can provide another perspective on the principles of phase separation in model systems [107,108,109,110].

### 4.1. Reconstitution of Biomolecular Condensates In Vitro

A variety of biomolecular condensates have been reconstituted in vitro: (1) to identify essential factors to form biomolecular condensates; (2) to test the systematic effects of external variables such as pH, salt concentration, temperature, and buffer composition; and (3) to characterize the biophysical features and material properties of the condensates. Typically, with dye-labeled recombinant proteins, RNA, or DNA, fluorescence microscopy can be used to monitor the behavior of individual biomolecules and condensates, owing to the high signal-to-noise ratio (Figure 3A). In addition, differential interference contrast (DIC) microscopy can be used to visualize biomolecular condensates of label-free biomolecules to avoid labeling artifacts. In addition, a DIC microscope can provide a higher contrast than a normal optical transmission microscope [111] (Figure 3B). Using these optical microscopes, solution-based biomolecular condensates can be reconstituted and visualized. For example, the interplay of proteins and DNA in in-vitro chromatin condensates was monitored by single-molecule fluorescence microscopy using immobilized fluorescence-stained DNA on the surface and labeled proteins [20,23,112,113] (Figure 3C). The use of DNA-staining fluorophores, such as YoYo1 or SYTOX Orange, enables labeled proteins to be monitored via single-molecule resolution to determine how many condensates are formed around the DNA and how the proteins induce topological changes in DNA. Moreover, AFM imaging, which provides high-contrast images, can be used to analyze biomolecular condensates in vitro [20,114,115] (Figure 3D). This enables the clear distinction of biomolecular condensates from individual proteins, RNA, or DNA, on an AFM microscope at sub-nanometer resolution.

### 4.2. Live-Cell Imaging

Live-cell imaging is vital for studying biomolecular condensates in a physiologically relevant context. Although in-vitro biomolecular condensates can provide quantitative and factorizable LLPS features, they should be tested under an in-vivo environment to provide biological context. Using live-cell imaging, we can study the material states of biomolecular condensates in a living cell by directly visualizing the condensates and monitoring the kinetics of fluorescence-labeled proteins (see Section 4.4).

Although early phase separation research focused on large biomolecular condensates in cells, such as HP1α condensates or P granules, using a normal optical microscope [11] (Figure 3E), recent studies have investigated smaller biomolecular clusters, such as ParB in bacterial cells, cohesin condensates, and transcription condensates [19,117]. Super-resolution microscopes, such as stimulated emission depletion microscopy (STED), photoactivated localization microscopy (PALM), and stochastic optical reconstruction microscopy (STORM), have been used widely to monitor small condensates that scale tens or hundreds of nanometers [19,118]. For example, the formation of 100 nm-sized transcriptional condensates was captured by PALM [19] (Figure 3F). In particular, super-resolution microscopes are essential for imaging biomolecular condensates in small prokaryotic cells.

An optogenetic protein construct was used to manipulate biomolecular condensates in living cells. The construct was oligomerized via laser excitation and fused with various interacting IDRs, such as FUS, DDX4, and hnRNAPA1 [105] (Figure 3G). A blue laser activated the oligomerization of the oligomerization domains (e.g., Cry2) and induced cytoplasmic and nuclear “optoDroplets” when the concentration of expressed constructs was sufficiently high (Figure 3H). At moderately supersaturated conditions above the threshold, FUS optoDroplets presented liquid-like properties, indicating that LLPS can be manipulated in a living cell [105]. The optoDroplet technique has also been used to draw a phase diagram “in cells,” which was consistent with that obtained in vitro [119].

### 4.3. Genomic Analysis

Chromosome conformational capture techniques, such as Hi-C, split-pool recognition of interactions by tag extension (SPRITE), tyramide signal amplification sequencing (TSA)-seq, and Hi-C chromatin immunoprecipitation (HiChipP), are widely used [56,57,58,60,61] to explain how LLPS is involved in the genome organization. For example, the genomic analysis was used to study chromosome compartmentalization probably induced by LLPS of HP1 or PolyComb [120,121].

In a typical Hi-C experiment, different chromatin regions that are in close spatial proximity are cross-linked, fragmented, ligated, and marked with adapters (Figure 3I). Fragments are then reverse cross-linked, purified, sequenced, and mapped to their genomic locations, yielding genome-wide contact frequency matrices (called the Hi-C map of compartmentalization). The segregation of heterochromatin and euchromatin can be easily observed by the checkerboard pattern of the Hi-C map (Figure 3J). In addition, techniques such as Chip-seq and ATAC-seq can be used to detect epigenetic marks or specific proteins involved in the phase separation of specific chromatin regions [67,68,69,70]. In particular, a combination of Hi-C and Chip-seq experiments has helped to determine how chromosome compartmentalization, at least partially induced by LLPS, can be linked to certain proteins, DNA sequences, and epigenetic marks [21].

### 4.4. Liquidity Test

Multiple experimental options can be utilized to determine the material states of biomolecular condensates in vitro and in vivo [122] (Figure 4). First, the shape of a condensate can reveal the liquidity of the droplet to some extent, because surface tension minimizes the surface-volume ratio by rearranging the molecules of the droplet (Figure 4A). To quantify the sphere-ness of a droplet, the *circularity*, defined as 4π*A*/*P*^2^, can be calculated by measuring the area of the droplet (*A*) and the perimeter of the droplet (*P*). The circularity is between 0 and 1 (perfect circle), depending on the closeness to a circle [20,24]. If two distinct droplets are fused to form a spherically reshaped droplet, this indicates that the droplet has liquidity that can rearrange molecules to minimize surface tension, as a single large sphere has a smaller surface-volume ratio than two smaller spheres (Figure 4B).

The mobility of individual molecules inside a droplet is a good indicator of liquidity. In a typical aqueous solution, liquid-like molecules diffuse much faster than solid- or gel-like molecules. Hence, molecules inside a liquid droplet are mobile, and the molecules are (relatively quickly) exchangeable between a droplet and the background solution. This mobility has been tested using FRAP experiments (Figure 4C). Using confocal microscopy, fluorescent molecules inside the small focal volume of a droplet are bleached, and the system is monitored to determine whether the bleached signals are recovered through the exchange of molecules between the bleached area and its surroundings.

1,6-hexanediol treatment is a typical method used to test the liquidity of condensates, since 1,6-hexanediol dissolves liquid droplets by inhibiting weakly hydrophobic interactions between molecules [123]. However, the results of recent studies suggested that 1,6-hexanediol treatment should be carefully considered when droplets are associated with chromatin, because the high concentration of 1,6-hexanediol can facilitate cation-dependent chromatin compaction [124,125]. Alcohols, such as 1,6-hexanediol, seem to remove water molecules around the chromatin and compact the chromatin [124]. Finally, reversibility is a common feature of liquid droplets. When background molecules are depleted, dissociation of a liquid droplet can be observed [20,23] (Figure 4D). These qualitative criteria can be used to determine the liquid-state condensations; however, quantitative analysis (such as viscoelasticity and hydrodynamics measurement [126]) is needed to define the exact material states of the biomolecular condensates, especially for in-vivo experiments.

### 4.5. Computational Modeling

Computer simulations have been adopted to provide a deeper understanding of the role of phase separation in chromosomes. In computer simulations, one can systematically alter models and parameters, which is limited and challenging in experiments, and this reveals the effects of different physical factors on the phase behavior of the modeled system and its consequences for the system properties, such as the chromosome structure.

Inspired by the polymeric nature of chromosomes, polymer simulations have been widely utilized to model chromosome systems. However, phase separation (and even microphase separation) is a collective behavior of particles and their interactions, which requires a large number of particles. Hence, to observe biomolecular phase separation *in silico*, a sizable system, and the corresponding computational costs, are inevitable. Therefore, atomistic simulations are rarely utilized to study phase behaviors [127].

A common strategy to overcome the system size problem is *coarse-graining*, which reduces the degree of freedom to describe each molecule [128,129]. Typically, a group of atoms is represented by a bead. For example, one can model each residue using a bead, and the whole protein becomes beads on a string. Although this (single-bead-per-residue) choice may seem natural, there is no golden rule for coarse-graining. As there is a tradeoff between resolution and computational cost, the details of coarse-graining depend on the system properties of the investigation. Coarse-grained models for biomolecular phase separation have been developed and deployed with a range of resolutions [35,37,47,74,130,131,132,133,134,135,136,137,138]. To further reduce the computational cost, the polymer system can be depicted by functional integrals over fluctuating fields; this is referred to as the *field-theoretic approach* [139]. This approach has recently been utilized to study biomolecular phase separation, especially when electrostatic interactions are dominant [140,141,142].

For chromosome modeling, the primary experimental target to reproduce computationally is often the Hi-C maps, as the contact information can be readily extracted from the simulation trajectories. As A/B compartmentalization is a notable feature of the maps, it can be reproduced by most simulations [143,144,145], and is usually explained by microphase separation [66,146]. Even the field-theoretic approach can reproduce A/B compartmentalization [147]. Another interesting topic is the role of phase separation at the level of the local structure of the chromosome; computational analyses were recently used to study various scenarios of local phase separation [148].

## 5. Local Phase Separation Models: BIPS and SIPS

Chromatin can undergo phase separation [149,150], and DNA-binding proteins can form liquid-like droplets around DNA in vitro and in vivo [19,20,83,84]. In addition, droplet formation can modulate the physicochemical properties of adjacent chromatin regions [105,151] through local changes in the effective interactions between different regions of the polymer, which induces a different microphase separation pattern. In the stickers-and-spacers framework, local phase separation can generate, modify, or remove stickers on chromatin. If phase separation locally gathers two distant chromatin regions by modulating their effective interaction strength, it can lead to a notable change in the chromatin structure. This mechanism is called bridging-induced phase separation (BIPS) [20] or polymer-polymer phase separation (PPPS) [152]. The hallmark of this model is that the mediating molecules do not undergo phase separation unless they are mixed with chromatin, which differentiates BIPS from typical phase separation [152].

### 5.1. BIPS versus SIPS

The BIPS model states that biomolecular condensates are formed via bridging of distant regions on a long DNA chain by proteins that possess multiple DNA-binding sites [20,152,153] (Figure 5A). Once a multivalent chromatin-binding protein connects two different DNA segments and forms a DNA loop, the bridged region can function as a nucleation point for further growth of condensates of the chromatin bridging proteins (Figure 5B). The BIPS model was initially suggested by molecular simulations, showing that a protein with more than two DNA-binding sites can be clustered along a DNA chain [108,154]. The clustering mechanism can be explained as follows: once a protein bridges two different DNA segments of a long DNA molecule to form a DNA loop, the local DNA concentration at the bridged region increases to recruit more DNA-binding proteins. In addition, the entropic loss (due to translational entropy of a DNA chain) is much lower when DNA-binding proteins bind to the bridged region of DNA than when they bind to a non-bridged region to form a new bridge [110,153].

The BIPS mechanism differs from typical phase separation that employs interactions between multivalent soluble proteins (Figure 5C), and some authors call this typical phase separation simply LLPS (as opposed to PPPS) [152,155]. However, since BIPS/PPPS can also induce liquid-like condensates [20], which is a hallmark of LLPS, we suggest that BIPS should be considered a type of LLPS. For non-BIPS LLPS, we propose a new term *self-association-induced phase separation* (SIPS). Note that LLPS implies that the resulting condensates possess liquidity, and SIPS can lead to the formation of gel- or even solid-like condensates [156], which may be confusing if we use LLPS instead of SIPS. Hence, we recommend that the use of LLPS be restricted to the formation of liquid-like condensates, regardless of the underlying molecular mechanism.

Although a protein in the BIPS model is involved in multiple DNA interactions, it does not require multiple protein-protein interactions, which are the main driving forces of SIPS. Thus, BIPS does not require an IDR of a scaffold protein, which typically provides multivalency and flexibility because flexible and long DNA can provide multiple binding sites for multivalent DNA-binding proteins. Moreover, while DNA organization is strongly coupled to DNA-protein cluster formation in BIPS, the organization of DNA can be completely independent of phase separation in SIPS (Figure 5D). Although the molecular mechanisms differ, BIPS shares many similarities with SIPS. For example, condensates formed by BIPS can have liquidity [20]. Hence, the techniques used to study SIPS can be applied to analyze BIPS.

### 5.2. Cohesin-Mediated BIPS

The cohesin-SMC complex is important for interphase chromosome organization [157,158], and in-vitro experiments have shown that the complex forms condensates via the BIPS mechanism [20]. Cohesin is a good model for a protein with multiple DNA-binding sites. Because it acts primarily as a motor protein to extrude a DNA loop for interphase chromosome organization, there are at least two DNA-binding sites on the surface of the cohesin protein for the relative motion of two different DNA-binding sites in an ATP hydrolysis-dependent manner. Multiple DNA-binding sites on the cohesin protein have been confirmed by various structural studies, suggesting that it can bridge distant DNA segments [158,159,160,161].

The cohesin-SMC complex has a non-monotonic size dependence on DNA length, and the cohesin-dependent BIPS mechanism can successfully explain the behavior by considering DNA bridging activity (Figure 5E,F) [20]. In an experiment, the DNA length was varied from 100 bp to 50 kbp, while the DNA concentration was fixed. The DNA-cohesin mixture was incubated and imaged using an AFM. For short DNA lengths (*l* < 3 kbp), no clear cohesin-DNA cluster was formed; however, beyond a crossover point of *l*_c_ ~ 3 kbp, the cluster size increased rapidly with DNA length, scaling as a power law (Figure 5F). The crossover point can be explained quantitatively by considering the free energy cost related to DNA looping by the bridging of a cohesin protein. When a single cohesin complex bridges two DNA sites to form a loop, the free energy change can be roughly estimated based on two contributions: (1) the DNA bending energy; and (2) the entropic cost due to DNA looping. The optimal length for DNA looping can be obtained by minimizing the following free energy:(8)$$\frac{F}{k_{B}T}=\frac{2\varepsilon l_{p}}{l}+1.5\log\left(\frac{l}{l_{p}}\right)$$
where *l* is the loop size when DNA is bridged by a single cohesin protein complex, *l_p_* = 50 nm is the persistence length of DNA, and *ε* = 16 is the shape parameter based on a tear drop [162]. The free energy is numerically minimized around the DNA length of 3 kbp, and hence, DNA must be at least 3 kbp to be bridged. A longer DNA construct (>3 kbp) provides a nucleation point for further growth of the condensates, which catalyzes cluster growth. The power-law scaling behavior of cluster size with DNA length was reproduced by computer simulations, which modeled cohesin as a patchy particle with two distinct DNA-binding sites [20].

### 5.3. Interplay of BIPS and SIPS

Although BIPS and SIPS seem to be opposing concepts, they can work together to induce efficient phase separation. As discussed, in the BIPS model, a bridged loop can act as a nucleation point (Figure 5B). The loop can attract multivalent proteins involved in SIPS (Figure 5C), resulting in the interplay between BIPS and SIPS. It is probable that some topologically associating domains (TADs), observed via Hi-C analysis [163], might be formed by BIPS, since an extruded DNA loop at the convergent CCCTC-binding factor (CTCF)-binding sites can act as a nucleation point for the growth of multivalent DNA-binding proteins assemblies. If this model is correct, interactions between a DNA loop and other nuclear condensates, such as transcriptional condensates, would be observed.

## 6. Concluding Remarks

In this review, we discuss the fundamental principles of biomolecular phase separation, including the stickers-and-spacers model, and the current understanding of phase separation involved in DNA-related processes in chromosomes. The stickers-and-spacers model is a simple conceptual framework inspired by polymer theories and can be applied to the phase separation of biopolymers with various architectures. Microphase separation is another important concept adopted from polymer physics, which can explain the segregation of euchromatin and heterochromatin. Various nuclear/chromosomal biomolecular assemblies formed by phase separation are involved in chromosome organization or many genome-related biologically important functions. Notably, in chromosomes, very long DNA molecules are involved in phase separation. The BIPS model illustrates the role of DNA in phase separation and utilizes multivalent DNA interactions of a protein to drive phase separation. However, although a few *in-silico* and in-vitro examples have been identified, it needs more examples of phase separation driven by proteins bridging two different DNA segments using multiple DNA-binding sites. Moreover, the in-vivo relevance of the BIPS model needs to be clearly demonstrated.

Currently, a complete conceptual framework to understand the phase separation involved in chromosome organization is lacking, as the inside of a cellular nucleus is filled with many molecules, forming a heterogeneous and disordered mass of biopolymers, which is far from being in equilibrium. To cut the Gordian knot of the chromosome, a fuller understanding of the local and global topologies of chromosomes is required. In particular, the static and dynamic actions of phase separation on DNA topology are key factors for determining the structure and properties of chromosomes, as illustrated by the BIPS model. We need to observe how topological changes in DNA induce phase separation in real time; however, the spatiotemporal resolution of current microscopes is limited. Monitoring the full dynamics of DNA topology and protein assembly may show this process clearly. Regarding simulations, although it is not yet possible to simulate the full nucleus on the atomic level, theoretical and numerical methods can provide valuable information that the experiment cannot access. In this complicated system, multi-scale simulations are promising for accessing a wide range of length and time scales. User-friendly and open-source packages will greatly benefit the community.

In the future, we expect that these approaches will provide a clear understanding of the role of phase separation in the chromosome, such as how the chromosome is segregated into two compartments, and how other bimolecular condensates involved in DNA-related functions are condensed with DNA. This understanding will help to explain multiple phenomena in the nucleus throughout the cell cycle.

## Figures and Tables

**Figure 1 ijms-22-10736-f001:**
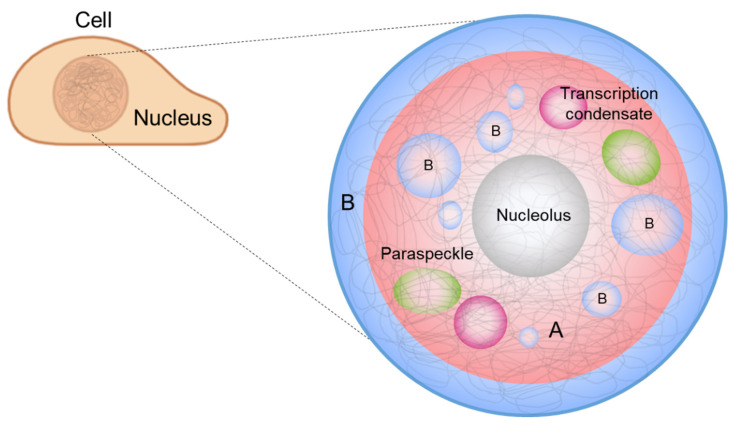
Biomolecular condensates in the nucleus: A and B compartments, nucleolus, paraspeckles, and transcriptional condensates. Chromosomes are largely segregated via phase separation into two compartments: euchromatin (A, red) and heterochromatin (B, blue). Phase separation is also involved in the formation and regulation of membraneless organelles such as the nucleolus (gray), transcription condensates (magenta), and paraspeckles (green) in the nucleus.

**Figure 2 ijms-22-10736-f002:**
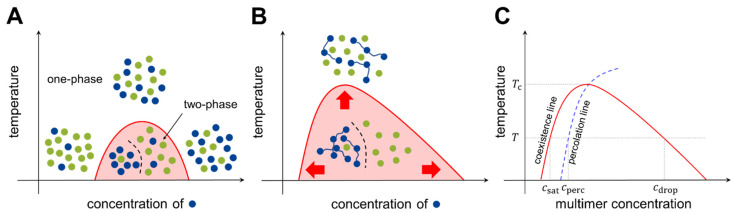
Phase diagrams of prototypical two-component systems. Phase diagrams for (**A**) the monomer-monomer system and (**B**) the polymer-monomer system. Blue and green dots represent different types of unit molecules. The *x*-axis indicates the concentration of unit molecules of the blue species, and the *y*-axis indicates the system temperature. In panel B, the valence of a multimer, *M,* is set to three. Multimerization results in the expansion of the two-phase regime. (**C**) Anatomy of a phase diagram (see text for the definitions of different concentrations). The *x*-axis shows the multimer concentration, and has a different scale from panels A and B. The multimer concentration, however, is proportional to the unit molecule concentration, and the two can be interchangeably used.

**Figure 3 ijms-22-10736-f003:**
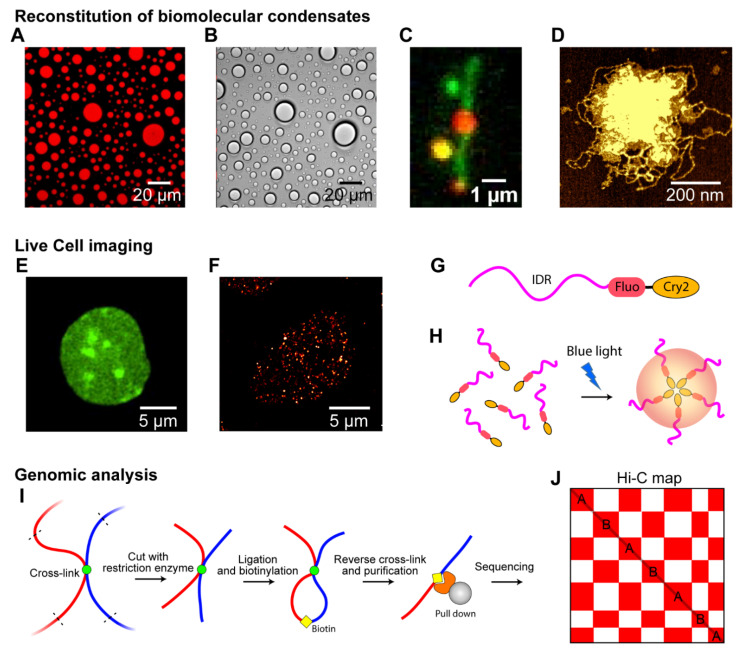
Techniques to observe phase separation in chromosomes. (**A**–**D**) Reconstitution of biomolecular condensates in vitro. (**A**) Fluorescence microscopy image showing liquid droplets of fluorescently labeled PRM-SH3-6His in the presence of Ni^2+^ ions obtained with a wide-field microscope (mCherry was fused with the protein) [38]. (**B**) DIC microscopy image showing unlabeled droplets of PRM-SH3-6His. (**C**) Single-molecule DNA tethered assay with fluorescent-stained DNA (green) and labeled cohesin proteins (red) forming condensates with DNA [20]. (**D**) AFM image of an unlabeled cohesin/DNA condensate [20]. (**E**–**H**) Live-cell imaging of biomolecular condensates. (**E**) Confocal microscopy image showing HP1α in the nucleus of a HCT116 cell. The condensates of Dendra2-tagged HP1α are clearly visible. (**F**) Super-resolution images of the Dendra2-Pol II cluster in a HCT116 cell obtained using a PALM microscope [19]. (**G**) Construct for “optoDroplet,” combining optogenetic-induced oligomerization with IDR-driven phase separation. The IDR (magenta) driving phase separation is fused with a fluorescent protein (red) and Cry2, a protein domain that forms oligomers upon blue-light activation [116]. (**H**) Blue-light activation induced the oligomerization of Cry2, which controls the interactions between IDRs to induce phase separation. (**I**,**J**) Genomic analysis via Hi-C. (**I**) Experimental scheme for Hi-C experiments. (**J**) Example genome-wide contact map (Hi-C map of compartmentalization). The X and Y axes denote the genomic positions in a chromosome. A and B compartments are shown in the Hi-C map by the “checkerboard” pattern. High frequencies of contacts are colored red, and low frequencies, white.

**Figure 4 ijms-22-10736-f004:**
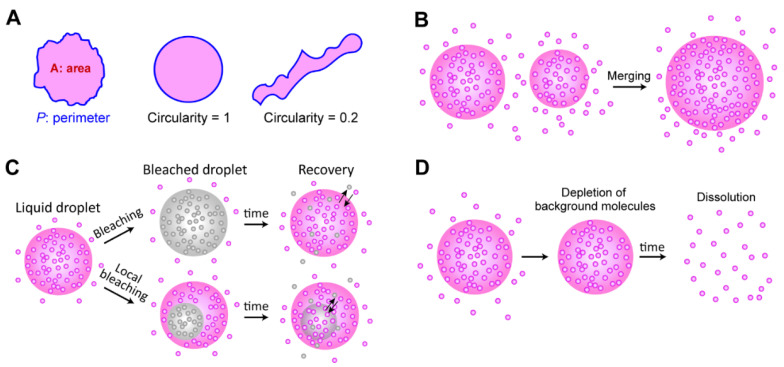
Criteria for liquidity. (**A**) The spherical shape of a liquid droplet. To quantify how much the droplet is close to a spherical shape, circularity is calculated by measuring the area (magenta regions) and the perimeter of a droplet (blue boundaries). The circularity is defined by 4π*A*/*P*^2^ and it ranges from 0 (very different from a circular shape) to 1 (a circular shape). (**B**) Merging of two distinct droplets. (**C**) FRAP experiment showing exchangeability of molecules between background solution and a droplet (top) and diffusability in a single droplet (bottom). (**D**) Reversibility test. If background proteins are depleted, a liquid droplet is dissolved into a solute phase.

**Figure 5 ijms-22-10736-f005:**
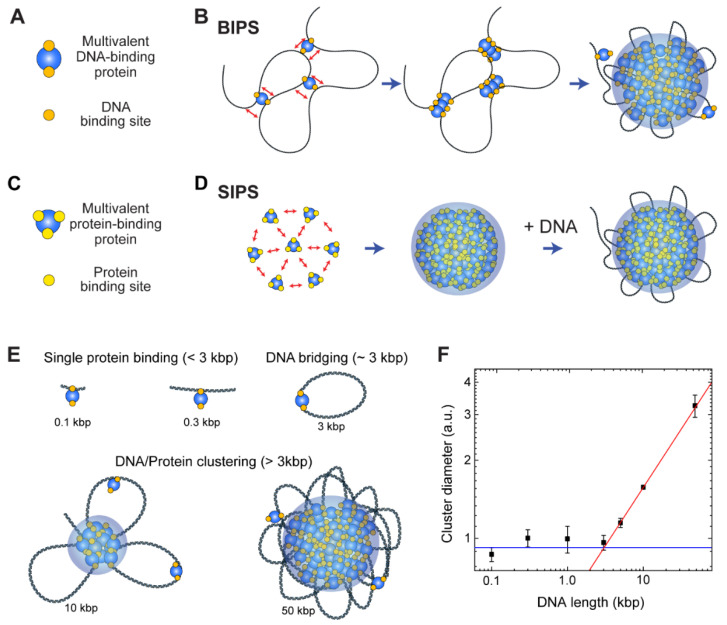
BIPS versus SIPS. (**A**) Cartoon of a multivalent DNA-binding protein that has at least two DNA-binding sites. DNA-binding sites of the protein are depicted as orange circles, and the protein is denoted as a blue circle. (**B**) Schematic of the BIPS model. Two DNA-binding sites per protein are sufficient for condensation, and a long DNA molecule is irreplaceable in this mechanism. (**C**) Cartoon of a multivalent protein-binding protein that induces typical phase separation. Yellow circles on the protein (blue circle) depict protein binding sites. (**D**) Typical phase separation mechanism (SIPS), which uses multivalent protein-protein interactions. At least three binding sites are necessary for phase separation, and DNA plays an auxiliary role in this process. (**E**,**F**) Dependence of the protein-DNA cluster size on the length of DNA shown in the previous study of cohesin-mediated BIPS [20]. (**E**) Cartoons of possible protein-DNA complex topologies for a range of DNA lengths and (**F**) a plot showing cluster size versus DNA length [20]. With <3 kbp of DNA, a single protein binds to DNA with no cooperativity (blue line). With ~3 kbp DNA, multivalent DNA-binding proteins can bridge a DNA to form a loop. For longer DNA (>3 kbp), a larger cluster can be formed, and the cluster size scales as a power law with the DNA length (red line).

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
