# Peer review of "Current Understanding of Molecular Phase Separation in Chromosomes"

_ijms, 2021, doi:10.3390/ijms221910736_

Round 1

Reviewer 1 Report

I found the review well organized, very interesting and pleasant to read. Concepts are well explained and examples easy to understand even for scientists without a strong background in this topic. The figures are clear and easy to understand, the bibliography is up to date.

Before publication, I have several points that the authors should consider. In particular, section 4 needs clarifications at different places (see Major points 11, 12 and 14). In my opinion, the reason "why" the methods described in the different subsections can be useful in the context of LLPS is not always clear. The description of the methods though is good, but they are not always replaced in the context of LLPS understanding/analysis.

Also, I found the part 2.2 a bit difficult to grasp, in particular why the authors insist on the difference between multi-domain proteins and intrinsically disordered proteins. This is not critical in my opinion, but rather a thought that the authors may want to consider. I find the main message of this part diluted.

Major points

M1: Page 1, line 31-32. A reference should be included for LLPS.

M2: Page 1-2, line 44-45. "membraneless organelles provide […] reaction centers." The sentence is a bit fuzzy because it is not very precise. I would recommend the authors to rewrite it so we can see the logical link between the two parts of the sentence (concentration heterogeneity and reaction center). A reference would be welcome here as well.

M3: Page 2, line 47. "[…] to an abrupt change". Change of what? Environmental change? Be more precise.

M4: Page 2, line 51-55. This part is missing references (reviews).

M5: Page 3, line 81-82. It is not clear to me what the "concentration" in panel A and B refers to. Even with the text, I cannot figure it out. Could the authors explain it? From the text, I understand that "concentration" in panel C refers to the concentration of the polymers. Correct?

M6: Page 3, line 87. Define in the text ΔFmix, as well as R and T (they are defined later, but one has to read further to find them).

M7: Page 4, line 128. Back to a previous comment, not clear to me what is the "system concentration". Even though this is obvious for the authors, I think this should be clearly written. The whole part (2.1) is well written, but without this information, I find it difficult to fully understand.

M8: Page 6, line 242-244. "[…] consistent with nucleosomes being sparsely distributed in euchromatin […]". I am not sure to understand the logic here. If euchromatin is more accessible, it should be more stained than heterochromatin, no? As far as I know, it is because heterochromatin is denser that it appears more stained (more material for the same volume). Or do accessibility and staining have nothing to do with each other? Either way, I do not see the consistency here.

M9: Page 8, line 318. Verify ref 82, it does not seem right to me.

M10: Page 8, line 320. "[…] replicate viral condensates". What does that mean?

M11: Page 9, Section 4.2. Live-cell imaging. It is not clear to me what information you can get from live-cell imaging in the context of LLPS. What can one measure with this method? Or is it just for observations?

M12: Page 9, Section 4.3. Genomic analysis. In the same vein as my previous comment, it is not clear how genomic analysis can be used to study LLPS phenomenon in vivo. What type of relevant information do these methods provide in the context of LLPS? One or two sentences should be enough.

M13: Page 11, line 452-453. "[…] quantitative analysis is needed to […]". What type of quantitative analysis? Of the same type of experiments?

M14: Page 11-12, Section 4.5. Computational modeling. As for M11 and M12, the reason "why" to use these methods in the context of LLPS is not clear. The authors describe well the principles behind the models used in such approaches. But "how" does it help studying LLPS?

Minor points

m1:  Page 1, line 28. Replace ";" with ",".

m2: Page 2, line 58. I would replace "nuclei" by "the nucleus".

m3: Page 7, line 260. Add "domains" (or something similar) after "chromatin"

m4: Page 7, line 274-275. "as related molecular components transition". I do not understand this part of the sentence.

m5: Page 7, line 294. "transcription factors (TFs) for RNA". Why "for RNA"?

m6: Page 9, line 379. Since only two references are indicated (original papers), I would remove "widely".

m7: Page 9, line 382. "bimolecular". Shouldn't it be "biomolecular"?

m8: Page 13, Section 5.2. Cohesin-mediated BIPS. Verify the use of "cohesion" instead of "cohesin".

m9: Page 15, line 610. "[…] involved in chromosomes […]". Should be "in chromosome organization", or something similar.

Reviewer 2 Report

The review article by Ryu et al. is well written and organized, with valuable information for experts and non-experts in the field of phase separation in the nucleus. The figures included nicely illustrate the main points addressed.

I only have a few minor suggestions:

-In the way it is currently written, section 5 may be a bit difficult to follow. While the introduction and sections 3 - 4 describe concepts, examples and experimental tools related mainly with liquid-liquid phase separation (LLPS), two kinds of phase separation models (BIPS and SIPS) are discussed in section 5. In this section, aside of LLPS, PPPS is also mentioned. For the sake of clarification, the authors have included a “small note on terminology” paragraph at the end of 5.1. I suggest shifting this explanation to the beginning of section 5 or, if possible, to the introduction, to make this point clear from the beginning. Related with this, in contrast with LLPS, PPPS and BIPS that correspond to terms regularly used in the field, SIPS seems less employed. The authors may consider replacing SIPS by a more familiar term describing this model, or citing a reference in which SIPS has been previously used, or proposing the use of SIPS.

-According to the main text, panels E and F in Figure 5 correspond to a particular example, the cohesion-SMC complex. In addition to citing the original reference, this should be specified in the figure legend.

-In figure 2, N and M may be defined in the legend. Moreover, to avoid confusion, M may be replaced by another letter/symbol, as it is used in the main text to indicate the number of binding units (valence) in each multimer.

-In figure 3, panel I, some of the symbols are not obvious and may be defined, for example the gray circle.

-The statement “prokaryotic nuclear proteins” (page 8, line 317) should be reconsidered, as prokaryotic cells do not have a defined nucleus but a nucleoid.

-On page 8, the heading of section 4.1 has been dislodged.

Round 2

Reviewer 1 Report

The authors replied very clearly to all my comments and improved, in my opinion, the readability and the quality of the parts I found unclear in the first version of the manuscript. I am taking this opportunity to thank the authors for a very nice review on the topic of LLPS, I enjoyed reading it.